# The Frequency of Tobacco Smoking and E-Cigarettes Use among Primary Health Care Patients—The Association between Anti-Tobacco Interventions and Smoking in Poland

**DOI:** 10.3390/ijerph191811584

**Published:** 2022-09-14

**Authors:** Małgorzata Znyk, Ilona Wężyk-Caba, Dorota Kaleta

**Affiliations:** Department of Hygiene and Epidemiology, Medical University of Lodz, Żeligowskiego 7/9, 90-752 Lodz, Poland

**Keywords:** counseling, e-cigarette, family doctor, minimal intervention, Poland, primary healthcare, smoking

## Abstract

The aim of this study was to assess the prevalence of smoking and e-cigarette use among primary care patients during the COVID-19 pandemic and to assess the frequency of minimal anti-tobacco interventions by family doctors. A cross-sectional study was conducted from January 2020 to December 2021 encompassing 896 patients over 18 years of age who used primary health care in the city of Lodz, Poland. In total, 21.2% of the respondents were smokers, 11.6% were e-cigarette users, and 7.3% dual users. In addition, 68.4% of smokers had been asked about smoking, while 62.9% of non-smokers and 33.7% of smokers were advised to quit smoking; furthermore, 71.1% of e-cigarette users and 72.3% of dual users were asked about tobacco use, and 17.3% and 21.5%, respectively, had been advised to quit smoking. Multivariate logistic regression analysis found men and alcohol users to receive more minimal anti-tobacco advice than women and non-alcohol users (OR = 1.46; *p* < 0.05 and OR = 1.45; *p* < 0.05), socio-demographic and health correlates did not increase the chances of obtaining minimal anti-tobacco interventions among smokers. People with a medium level of education had a higher chance of receiving minimal anti-tobacco intervention from their family doctor when using e-cigarettes and when they were dual users (OR = 2.06; *p* < 0.05 and OR = 2.51; *p* < 0.05). Smokers were less likely to receive minimal anti-tobacco interventions than reported in previous studies. Measures should be implemented to increase the minimum interventions provided by GPs in their daily work among all patients, not only those who use tobacco. Non-smokers should be encouraged to abstain.

## 1. Introduction

Globally, tobacco kills more than 8 million people each year, of which more than 7 million are due to direct tobacco use [1]. Smoking is also responsible for the loss of 2,060,000 years of healthy life, which accounts for 16.3% of the total DALY value (disability-adjusted life-year) [2].

Smoking is a risk factor for premature mortality associated with common chronic diseases, such as cardiovascular disease, cancer, diabetes, and chronic respiratory diseases [3]. In 2020, 22.3% of the world’s population smoked tobacco: 36.7% of all men and 7.8% of women (World Health Organization data) [1].

Together with alcohol consumption and behavior leading to overweight and obesity, smoking is a major public health problem in Europe [3]. Nearly 27% of the population in the WHO European Region use tobacco, and more than 35% of men [4]. Tobacco consumption is also a public health concern in Poland [5], where in 2018, 20.2% of women and 30.9% of men smoked cigarettes; of these, the percentage for women was slightly below the mean European Union value (20.8%) while the men were slightly above (28.1%) [2]. In 2019, 21% of Polish citizens reported smoking addiction [6].

Electronic cigarettes have also gained popularity in recent years and are currently used by approximately 303 million people aged 15 and over globally [5,7]. In many cases, individuals report joint tobacco and e-cigarette use.

Due to the harm that smoking causes to both the health of smokers and those around them, many smokers attempt to quit smoking; however, only 4% who try to quit are successful [1]. Failure to stop smoking may be due to symptoms of nicotine withdrawal [8], which may also increase the likelihood of resuming smoking [9].

As recommended by the Centers for Disease Control and Prevention (CDC), anti-tobacco counseling can be delivered in a variety of ways, such as in-person by a healthcare professional, either one-on-one or in a group, and over the telephone through a quitline. Adults who smoke should talk to their doctor about proven smoking cessation methods, such as counseling and FDA-approved (Food and Drug Administration) medications. E-cigarettes are not currently approved by the FDA for helping relieve nicotine addiction [10].

Healthcare professionals often advise patients to improve their health through minimal interventions [11]. The activities of family doctors in primary health care may play a significant role in reducing the prevalence of smoking in Poland [12].

A motivational interview (MI) by physicians is a patient-centered style of counseling designed to help people explore and resolve ambivalent behavioral changes [13]. The aim of MAI (Minimal Anti-Tobacco Intervention) provided as part of medical advice by a primary care physician is to identify a cigarette smoker and present routes out of addiction [12]. MAI is based on the 5 × P strategy (translated as *ask, advise, plan, help*, *remember*). In each patient, the physician should assess the degree of addiction (e.g., Fagerstrom’s test) and readiness to quit (e.g., Shneider’s test) [14].

Primary care physicians should try to improve their qualifications in this area, which will increase the effectiveness of MAI. Those who train primary care physicians have a responsibility to ensure that smoking cessation is given appropriate emphasis in the curriculum.

However, in Poland, there is room for improvement regarding prevention in primary health care. According to the new regulations, every primary care patient has the right to coordinated care, intended to improve in the availability, quality, and effectiveness of care, as well as the level of patient satisfaction (The primary health care unit (POZ) is a part of the health care system that provides comprehensive health care for insured persons who have declared their willingness to use the services of their family doctor. The family doctor plans and implements medical care and coordinates the provision of services by the medical personnel cooperating with him). They recommend that in the first stage, the coordinator should focus on prevention [15]

The COVID-19 pandemic has affected patient lifestyles and has resulted in the coexistence of negative health behaviors. Various studies have been conducted on lifestyle changes during the pandemic, including eating habits, physical activity habits, smoking habits, and alcohol consumption [16,17,18,19,20,21]. Increased time at home has led to excessive food consumption and changes in physical activity, and an increased prevalence of overweight and obesity [22,23].

The period has also been characterized by changes in smoking habits. Some studies indicate a greater frequency of smoking [18,19,20], while others report a lower frequency, with greater intention and more attempts to quit smoking [21,24]. Others report a higher level of smoking to reduce loneliness at home or stress during the pandemic [25,26], and an increased number of smokers who tried to quit [27].

During the COVID-19 pandemic in Poland, a significant percentage of people (both men and women) decided to change their smoking habits (40.8% men and 31.2% women); however, men were more likely to make unfavorable decisions about smoking, e.g., to start smoking or smoke more often, than women (23.1% of men and 16.4% of women) [2].

The aim of the study was to assess the prevalence of smoking among primary care patients during the COVID-19 pandemic and to determine the frequency of minimal anti-tobacco interventions by family doctors in patients.

## 2. Materials and Methods

### 2.1. Study Design and Population

From January 2020 to December 2021, a cross-sectional study was conducted among 896 adult primary care patients in Lodz, Poland. From the list of 211 primary health care facilities in the city of Lodz, every fifth clinic was selected randomly. Thirty-four primary healthcare facilities agreed to conduct the study among patients. Every fifth patient leaving the doctor’s office was randomly selected and asked to participate in the study. Inclusion criteria: people over 18 years of age who consulted a doctor in primary care and agreed to participate in the study. The methodology, the study sample and the examined region are given elsewhere [28]. The study design was approved by the Bioethics Committee of the Medical University of Lodz (on 18 September 2018; RNN/315/18/ KE). Informed consent was obtained from all the study respondents.

### 2.2. Study Variables

The research tool was an anonymous questionnaire containing standardized questions that had been used in other studies [29,30]. Detailed information on smoking and e-cigarette use and anti-tobacco counseling was obtained through a questionnaire completed during a face-to-face interview conducted by the principal researcher.

The questionnaire included questions about socio-demographic characteristics (gender, age, marital status, employment status, and education), tobacco smoking and e-cigarette use, information about visits to family doctors, and questions about anti-tobacco counseling provided by GPs. This present article covers four of the seven sections included in the questionnaire: information on tobacco smoking, passive exposure to tobacco smoke, e-cigarette use, and information about appointments with a primary healthcare doctor.

The current smoking status was assessed based on the question “Do you currently smoke cigarettes?” Subjects who answered “no” were classified as compliant with smoking recommendations [31]. A daily tobacco smoker is defined as a person who smokes at least one cigarette every day during a period of 30 days. An occasional smoker is someone who has smoked less than one cigarette per day in the last 30 days.

The current use of e-cigarettes was assessed by the question “Do you currently use e-cigarettes?”. Daily e-cigarette users are people who have used an e-cigarette at least once per day for the last 30 days. Those who qualified as both cigarette users and e-cigarette users were defined as dual users.

Information on minimal anti-tobacco counseling provided by a family doctor was obtained based on the following survey questions: ‘’How often does your family doctor give you advice about smoking/ e-cigarette use”. People who have never been counseled on anti-tobacco smoking are respondents who answered “never”, and those who answered “always”, “often,” or “sometimes”, were classified as patients who received minimal counseling on tobacco smoking.

The answer “always” related to advice given at each visit to a family doctor. The answer “sometimes” indicated less than 50% of all medical appointments in primary care, and “often” as 50% or more medical appointments in primary care [28].

Participants were divided into four groups based on questions about chronic diseases treated by a GP: no disease, one, two, and three or more diseases. The participants were also asked about alcohol consumption (Table 1).

Univariate and multivariate analyses were performed regarding the minimum anti-tobacco intervention provided to (1) all patients, (2) smokers, (3) e-cigarette users, and (4) dual users.

### 2.3. Statistical Analysis

The results are presented as numbers and percentage rates. Descriptive statistics were used, and the analyzed variables were distributed. Single- and multivariable logistic regression analyses were performed to obtain OR (odds ratio) and 95% confidence interval (CI) for each anti-tobacco advice indicator. Variables with a *p* value of 0.1 or less from the univariate analysis were included in the multivariate model. A *p* value < 0.05 was considered statistically significant. STATISTICA software, version 13.3 (StatSoft, Tulsa, OK, USA), was used for the calculations.

## 3. Results

### 3.1. Characteristics of the Studied Population

The characteristics of the studied population of primary care patients are presented in Table 1.

The majority of the respondents were people with secondary education (56.9%), professionally active (61.6%), aged <30 (28.6%), and aged 40–49 (24.0%). Women constituted 74.2% of the respondents, and men 25.8%.

In addition, 81.9% of participants admitted to drinking alcohol, 22.9% were being treated for one chronic disease, 18% for three or more chronic diseases. The response rate of participants was high compared to other surveys in Poland (80%). There was no lack of data in the responses included in the analysis.

### 3.2. Smoking Tobacco

In total, 21.2% of the respondents were smokers. There were no occasional smokers.

The prevalence of daily tobacco smoking is presented in Figure 1.

The largest group consisted of smokers who smoked 5–10 cigarettes a day. Among the smokers, 56.8% smoked filtered cigarettes and 24.7% menthol cigarettes (Table 2). In addition, 37.9% of smokers smoke their first cigarette within 30 min after waking up, 33.2% smoke cigarettes during illness that makes them stay in bed, and 22.6% of smokers find it difficult to refrain from smoking in non-smoking areas. In addition, 35.8% of respondents answered that the most difficult thing was to give up the first cigarette. In total, 58.9% of daily smokers have attempted to quit, and 62.5% of respondents made two to five attempts to quit smoking. More than half (53.6%) of the respondents made the last attempt to quit smoking for 24 h more than one year previously. Finally, 56.8% of respondents reported smoking tobacco for one to ten years.

It was also found that 42% of respondents are exposed to secondhand smoke, most often at home (16.0%) and in other situations (17.8%). Participants staying in rooms where someone smokes tobacco usually stay there for less than one hour during the day (24.6%).

### 3.3. Use of E-Cigarettes and Dual-Use

In total, 28.6% of respondents indicated that they had ever used e-cigarettes, 5.9% of all participants use e-cigarettes every day and 5.7% use e-cigarettes occasionally. The prevalence of daily use of e-cigarettes is presented in Figure 2.

The respondents most often used e-cigarettes containing nicotine (83.7%). E-cigarettes were most often used once per day (40.4%). In addition, 7.3% of participants were dual users.

### 3.4. Minimal Anti-Tobacco Intervention in Primary Health Care

A total of 36% of respondents had never been asked about smoking during their GP visit, and 31.6% of smoking patients and 37.1% of non-smoking patients were never asked about smoking (Table 3). Among smoking patients, 68.4% were asked about smoking, and 62.9% among non-smokers.

Among the smoking subjects, 33.7% had been advised to quit smoking, most commonly in the last three years or more. It was also reported that GPs had presented methods and ways of quitting smoking or presented appropriate materials to 21.1% of smokers and 1.3% of non-smokers.

In addition, 34.2% of smoking patients were asked about the number of cigarettes smoked, and 27.4% had been advised to reduce the number of cigarettes smoked during the day. In addition, 27.9% of smokers reported that the GPs wrote in their documentation that they smoked cigarettes, and 42.1% of smokers were not interested in registering.

Our findings indicate that 34.7% of smokers and 11% of non-smokers were informed about the negative effects of smoking and the health consequences of smoking. In addition, 29.5% of smokers and 6.7% of non-smokers were informed by the GP about the negative impact of smoking during treatment for other diseases.

It was found that 28.9% of e-cigarette users and 27.7% of dual users had never been asked about smoking; however, 71.1% of e-cigarette users and 72.3% of dual users were asked about smoking (Table 4). In addition, 17.3% of e-cigarette users and 21.5% of dual users were advised to quit smoking. The GPs presented different methods and ways of quitting smoking or presented appropriate materials to 21.1% of e-cigarette users and 13.8% of dual users. Among these groups, 19.2% of e-cigarette users and 27.7% of dual users were asked about the amount of e-cigarettes used or cigarettes smoked, and 16.3% of e-cigarette users and 20% of dual users were recommended to reduce the number of cigarettes or e-cigarettes smoked during the day.

The health care physician recorded e-cigarette or tobacco use in the medical records of 17.3% of e-cigarette users and 18.5% of dual users. In addition, 27.9% of e-cigarette users and 32.3% of dual users were informed about the negative effects of smoking, health consequences, and the possibility of smoking-related diseases, while 18.3% of e-cigarette users and 20% of dual users were informed about the negative effects on the treatment of other diseases.

The relationships between personal characteristics (sex, age, education, professional status, marital status), total minimum anti-tobacco intervention and minimum anti-tobacco intervention for smokers, e-cigarette users, and dual users were investigated using logistic regression analysis. The strengths of any associations were determined using OR (odds ratio) and the 95% confidence interval (Cl). The results of the univariate and multivariate logistic regression analysis for general practitioner MAI with health and socio-demographic correlates are given in Table 5 and Table 6. Any variables that were found to be significant in the univariate logistic regression analysis were included in a multivariate logistic regression analysis.

Age 50–59 years (*p* < 0.05), primary education (*p* < 0.05) and secondary education (*p* < 0.001), and three or more chronic diseases (*p* < 0.05) were found to be statistically significant in the univariate analysis but statistically insignificant in the multivariate analysis.

In the multivariate logistic regression analysis, male participants and alcohol consumers received minimal anti-tobacco advice more often than female and non-alcoholic consumers (OR = 1.46; *p* < 0.05 and OR = 1.45; *p* < 0.05). In the case of smokers, primary education (*p* < 0.05) and secondary education (*p* < 0.01), and two chronic diseases (*p* < 0.05) were found to be statistically significant in the univariate analysis. The results are presented in Table 5. The multivariate analysis did not indicate that socio-demographic or health correlates increased the chances of obtaining minimal anti-tobacco interventions among smokers.

Moreover, the relationship between personal characteristics and the minimal anti-tobacco intervention given to e-cigarette users and the minimal anti-tobacco intervention given to dual users was examined (Table 6). For e-cigarette users, variables such as age 30–39 (*p* < 0.05), 40–49 (*p* < 0.05), 60+ (*p* < 0.001), single (*p* < 0.01) and pensioner (*p* < 0.001) were statistically significant in the univariate analysis but not in the multivariate logistic regression analysis. In the case of dual users, age 60+, single, unemployed, and student were statistically significant in the univariate analysis (*p* < 0.05), but not in the multivariate analysis.

The multivariate analysis found that people with medium education were twice as likely to receive minimal anti-tobacco intervention when using e-cigarettes and when they were dual users (OR = 2.06; *p* < 0.05 and OR = 2.51; *p* < 0.05).

## 4. Discussion

Our cross-sectional study is one of the first in Poland to address the subject of minimal anti-tobacco intervention by family doctors in primary care during COVID-19. It was found that 21% of the respondents smoke tobacco: 22.1% of the women and 18.6% of the men. These results are higher than those obtained in another study in Poland in 2020 [2], which found that 14.9% of women and 23.1% of men smoked, but lower than those obtained in 2018 (23.1% and 27.8%). In addition, 5.9% declared e-cigarette use (5.9% of women and 6.1% of men). A similar study in Poland yielded higher results: e-cigarette use was declared by 7.1% of women and 10.8% of men, of whom 4.1% of women and 7.0% of men were non-smokers [2].

The mean prevalence of smoking in the EU is 20.8% for women and 28.1% for men, with women ranging from 13.8% (Romania) to 30.0% (Croatia) and men from 18.5% (Denmark) to 49.1% (Cyprus) [32]. In addition, the percentage of smoking in Poland is slightly below the mean European Union value for women, and slightly above the value for men (20.2% for women in Poland and 30.9% for men) [32]. Smoking rates are higher in some regions of Asia (China and India) but are lower in Australia and North America [33]. In Poland, 1% of the population are current e-cigarette users (for EU + UK 2%) [34].

Various guidelines and protocols for healthcare professionals have been developed to help quit smoking. The minimum anti-tobacco intervention recommended in Poland is based on the WHO recommendations for the treatment of tobacco dependence and the guidelines adopted by the American Medical Association (AMA) [12].

According to the guidelines of the College of Family Physicians in Poland, a family doctor should conduct a minimal anti-tobacco intervention with each smoking patient at least once a year and record it in the patient’s documentation [12]. Due to the low effectiveness of a single MAI, an interview with a smoking patient should be conducted at each visit [35]. In its modified form, the MAI should also be used for non-smoking patients who are sometimes passively exposed to smoking. Non-smokers should be encouraged to abstain [12].

Our study found a low percentage of GPs administering minimal tobacco control interventions. Only every second person who joined our study admitted to having received MAI advice from their GP; this advice was offered sometimes, i.e., not at every routine visit. Only every twentieth respondent indicated that the doctor asked about smoking during each routine visit. Smokers were advised more often than non-smokers. Our data confirm previous findings indicating that MAI advice is given infrequently [36,37,38].

If the patient wants to quit smoking, the family doctor should provide advice, covering all available treatments or other supportive methods of therapy [12]. In our study, 64% of respondents received brief advice as a part of minimal intervention, while 10% received an intensive intervention. Other studies conducted in Europe (England, Germany, Greece, Hungary, the Netherlands, Poland, Romania, and Spain) and the United States have also noted that primary care patients reported a small percentage of minimal tobacco control counseling [39,40,41]. These findings indicate that anti-tobacco counseling was provided at a low level before the COVID-19 pandemic, in developed and developing countries. It is unclear whether COVID-19 was the cause of consultation.

However, higher minimal intervention rates were found elsewhere, where more than 2/3 of patients were asked about tobacco use, and about half were advised to quit (Indie) [42].

Research shows that motivational interviews (MI) produce a small but significant increase in smoking cessation compared to quick advice. In addition, MIs conducted by the primary care physician were more effective at quitting smoking than those provided by counselors or nurses. Shorter motivational conversation sessions of less than 20 min were more effective than longer ones [13]. Even three-minute tips can improve quitting rates [43,44]. However, studies have found MIs to be more effective than short quitting advice [45,46,47,48].

Various meta-analyses have found that quick advice increases the number of attempts to quit smoking by 1% to 3% [11], while short advice showed a significant increase in the rate of smoking cessation compared to no advice [11,47,49,50,51]. In addition, brief counseling and MI have comparable results for quitting smoking [11,47,52], and minimal tobacco control interventions (e.g., short advice) are less effective than behavioral counseling [53,54].

Guidelines also indicate that more intensive counseling (≥20 min per session) is more effective than less intensive counseling (<20 min per session) [55]. A more complex MI is needed when a healthcare professional encounters people with low motivation to change behavior [56]. Even so, with lifestyle modifications, a motivational conversation can be more effective than simply trying to change smoking habits [57]. Either way, MAI increases the chance of quitting smoking versus no such intervention [58,59].

Unlike previous studies, our present study evaluated the effect of minimal anti-tobacco interventions (MAI) given to e-cigarette users and dual users. The e-cigarette is a new form of nicotine delivery for people switching from flammable cigarettes [60,61]. In some countries (e.g., the United Kingdom), e-cigarettes are used as an aid to stop smoking [33]. Our findings indicate that e-cigarette users and dual users were more likely to receive advice than smokers, as noted previously [34,56]. Studies performed in other countries found that few smokers received advice on e-cigarettes from health professionals [41].

The inclusion of 5As intervention in primary health care increases patient satisfaction with the services provided by doctors. Satisfaction with counseling services can increase the willingness to quit smoking among primary care patients [62].

Our study found that the likelihood of receiving counseling was influenced by sex and alcohol consumption. GPs were more likely to advise when the patient was male and consumed alcohol, as in other studies [63]. The results reflect the fact that a number of preventive programs in Poland (e.g., cardiovascular diseases, metabolic diseases) are more often directed toward men [36,64]. Prophylaxis should be provided to patients at the same level, regardless of sex.

Quitting can reduce the chance of chronic diseases caused by smoking, including diseases of the respiratory system and cardiovascular systems. This is all the more important in the time of the COVID-19 pandemic, as it increases the risk of cardiovascular complications and other serious complications related to COVID-19 in smokers and vapers [65,66].

Our study found that GPs do not generally explain to patients that tobacco use has a negative effect on any chronic diseases they may have, with only one in every ten primary care patients being informed. Informing patients about the negative effects of smoking during the COVID-19 pandemic may have influenced their attempt to quit smoking; messages linking COVID-19 with smoking and the potential damage may hold promise for discouraging smoking and vaping [67].

Our findings indicate a lack of education in primary care facilities, which represent the first point of contact for patients. They also demonstrate that proper training of staff in this area can have a considerable overall impact.

In Poland, preventive actions are generally scattered and uncoordinated, and not only during the COVID-19 pandemic. They lack a clear strategy regarding their structure and direction. Preventive measures are underfunded, and their organization is generally poor; furthermore, the staff is used ineffectively, and there is no evaluation of preventive activities. In addition, at the level of family doctors in primary care, anti-tobacco interventions are underfinanced and are carried out without attention or proper implementation. While it is recommended that a minimum of EUR 2 per capita per year be spent on tobacco control, the 2018 Tobacco Control Budget (TCB) in Poland allocated EUR 0.006 per capita [68].

The persistent high prevalence of smoking in Poland, along with the increasing use of e-cigarettes and the low level of anti-smoking advice provided, should be of concern to both the government and medical organizations. The situation requires strong and comprehensive action by government and health authorities to implement the comprehensive approach recommended by the WHO; this includes measures set out in the WHO Framework Convention on Tobacco Control (FCTC) including Article 14, which outlines best practice approaches to smoking cessation, together with other measures needed to reduce smoking in the community, such as strong curbs on tobacco industry marketing and other activity, strong health warnings, price/tax policy and public education [69]. In line with the provisions of the FCTC, each country should promote and strengthen public awareness of tobacco control issues (Article 12) [70]. Effective action on tobacco by governments around the world has generally been prompted by calls for action by medical and health organizations; therefore, there may well be a role for such approaches in Poland.

In Poland, due to poor enforcement of the earmarking law, the 2000–2018 Tobacco Control Program was beset by inadequate planning and insufficient financing. This should be regarded as a warning to other countries to create legislation that can be verified and controlled [70].

The study has limitations. First, as it is a cross-sectional study performed at a single point in time, changes cannot be seen over longer periods; one-time minimal anti-tobacco intervention is insufficient, it must be carried out over a long period. Second, the study was conducted during the COVID-19 pandemic, which complicated access to patients in primary care, and not all patients visited their GP during this period. Finally, minimal anti-tobacco intervention by a GP was assessed based on patient questionnaire data; as such, a bias associated with recall may exist. Furthermore, the questionnaire did not address the reason for the patient visiting the doctor; however, minimal anti-tobacco intervention should be provided at least once per year during such visits, regardless. In addition, the small size of the sample could have affected the lack of association of variables in multivariate analyses. Finally, the study was anonymous, and hence, it was not possible to connect the study participants with their family doctors.

Nevertheless, this study is the first to evaluate the conduct of MIA in Poland by family doctors during the COVID-19 pandemic. It also examines many determinants that may influence primary health care counseling in the study population. In addition, it describes an urban population, which allows for further generalization of the results to other urban areas and populations.

Our findings may be important for the development of preventive measures aimed at reducing tobacco consumption in Poland and other countries.

## 5. Conclusions

Higher levels of counseling were found to be associated with male sex and alcohol consumption. The COVID-19 pandemic was a difficult period that contributed to a rise in harmful health behaviors. The study confirms that the frequency of minimal anti-tobacco interventions by family doctors is not sufficient: GPs in primary care appear to treat patients rather than prevent disease. Prophylaxis is not working, which was further demonstrated by the COVID-19 pandemic. Family doctors should take steps to educate and promote a healthy lifestyle among patients in order to increase positive pro-health behaviors, particularly during the COVID-19 pandemic. During routine visits, the family doctor has the opportunity to constantly motivate and prepare the patient to quit smoking.

Minimal anti-tobacco interventions should be provided to all patients, not only smokers. Non-smokers should also be encouraged to abstain.

Early prophylaxis can provide more benefits than modifying the treatment, and such strategies can play an important role in improving the health of the population.

## Figures and Tables

**Figure 1 ijerph-19-11584-f001:**
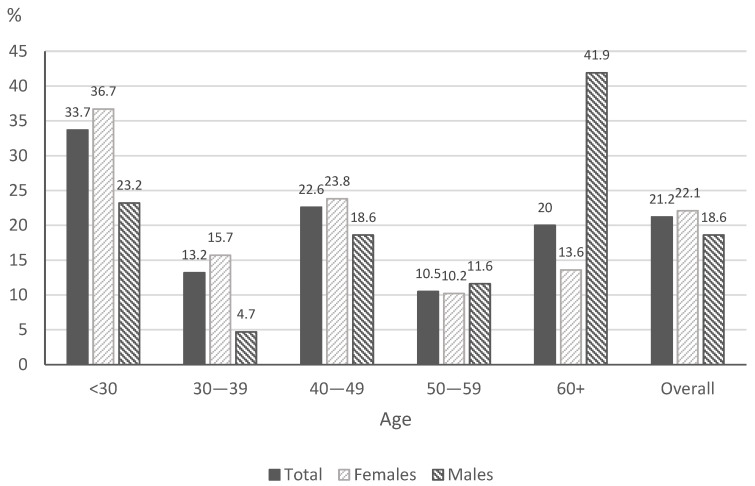
The prevalence of daily tobacco smoking.

**Figure 2 ijerph-19-11584-f002:**
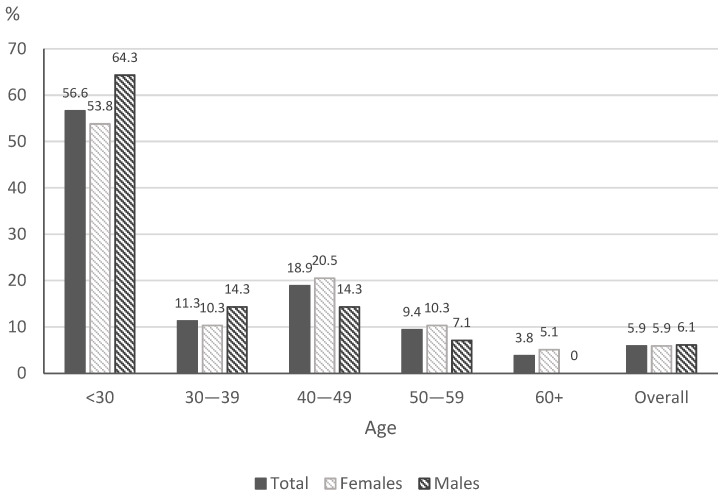
The prevalence of daily use of e-cigarettes.

**Table 1 ijerph-19-11584-t001:** Characteristics of the studied population.

Characteristics	Total *n* = 896	%
**Gender**		
Female	665	74.2
Male	231	25.8
**Age (years)**		
<30	256	28.6
30–39	123	13.7
40–49	215	24.0
50–59	105	11.7
60+	197	22.0
**Education**		
Primary	26	2.9
Medium/Secondary	510	56.9
Post-secondary vocational	74	8.3
Higher	286	31.9
**Marital status**		
Single	394	44.0
Married	374	41.7
Widowed	69	7.7
Divorced	59	6.6
**Professional situation**		
Unemployed	45	5.0
Professionally active	552	61.6
Pensioner	144	16.1
Student/pupil	155	17.3
**Number of chronic diseases**		
0	420	46.9
1	205	22.9
2	109	12.2
≥3	162	18.0
**Alcohol consumption**		
Yes	734	81.9
No	162	18.1

**Table 2 ijerph-19-11584-t002:** Tobacco smoking prevalence among primary care patients.

Characteristics	Total *n* = 896 (%)	95% CI	*p* Value
**Tobacco smoking**			
Yes	190 (21.2)	(18.5–23.9)	0.2121
No	706 (78.8)	(76.1–81.5)	0.7879
**Number of cigarettes smoked per day**			
<5	52 (27.4)	(21.0–33.7)	0.2737
10 May	76 (40.0)	(33.0–47.0)	0.4
20 November	58 (30.5)	(24.0–37.1)	0.3053
>20	4 (2.1)	(0.06–4.1)	0.0211 *
**The type of cigarettes smoked**			
With filter	108 (56.8)	(49.8–63.9)	0.5684
Without filter	3 (1.6)	(−0.19–3.35)	0.0158 *
Hand rolled	6 (3.2)	(0.7–5.6)	0.0316 *
Slim	26 (13.7)	(8.8–18.6)	0.1368
Menthol	47 (24.7)	(18.6–30.9)	0.2474
**Try to quit smoking**			
Yes	112 (58.9)	(52.0–66.0)	0.5895
No	78 (41.1)	(34.1–48.0)	0.4105
**Number of attempts to quit smoking**			
1	29 (25.9)	(17.8–34.0)	0.2589
5 February	70 (62.5)	(53.5–71.5)	0.625
10 June	9 (8.0)	(3.0–13.1)	0.0804
>10	4 (3.6)	0.13–7.0	0.0357 *
**The last attempt to quit smoking for 24 h**			
In the last month	23 (20.5)	(13.1–28.0)	0.2054
More than 1 month to half a year ago	12 (10.7)	(5.0–16.4)	0.1071
More than half a year to 1 year ago	17 (15.2)	(8.5–21.8)	0.1518
Over 1 year ago	60 (53.6)	(44.3–62.8)	0.5357
**How many years have you been smoking addicted**			
<1	2 (1.1)	(−0.4–2.5)	0.0105 *
10 January	108 (56.8)	(49.8–63.9)	0.5684
20 November	42 (22.1)	(16.2–28.0)	0.2211
>20	38 (20.0)	(14.3–25.7)	0.2
**How soon after you wake up do you smoke your first cigarette?**			
within 30 min			
after 30 min	72 (37.9)	(31.0–44.8)	0.3789
	118 (62.1)	(55.2–69.0)	0.6211
**Do you find it difficult to refrain from smoking in non-smoking areas?**			
Yes			
No	43 (22.6)	(16.7–28.6)	0.2263
	147 (77.4)	(71.4–83.3)	0.7737
**Do you smoke when you are ill, which forces you to stay in bed most of the day?**			
Yes			
No	63 (33.2)	(26.5–39.9)	0.3316
	127 (66.8)	(60.1–73.5)	0.6684
**Which cigarette would be the most difficult for you to give up?**			
from the first			
everyone else	68 (35.8)	(29.0–42.6)	0.3579
	122 (64.2)	(57.4–71.0)	0.6421
**Are you exposed to secondhand smoke?**			
Yes, only at home	143 (16.0)	(13.6–18.4)	0.1596
Yes, only at work	37 (4.1)	(2.8–5.4)	0.0413 *
Yes, at home and work	37 (4.1)	(2.8–5.4)	0.0413 *
Yes, in other situations	159 (17.8)	(15.2–20.2)	0.1775
Not	520 (58.0)	(54.8–61.3)	0.5803
**How many hours a day do you spend in rooms where someone smokes tobacco?**			
I’m not in such rooms at all			
less than 1 h during the day	607 (67.7)	(64.7–70.8)	0.6775
from 1 h to 5 h during the day	220 (24.6)	(21.7–27.4)	0.2455
5 to 8 h a day	53 (5.9)	(4.4–7.5)	0.0592
	16 (1.8)	(0.9–2.7)	0.0179 *
**Have you ever used e-cigarettes?**			
Yes	256 (28.6)	(25.6–31.5)	0.2857
No	640 (71.4)	(68.5–74.4)	0.7143
**Do you currently use e-cigarettes?**			
Yes, daily	53 (5.9)	(4.4–7.5)	0.0592
Yes, occasionally	51 (5.7)	(4.2–7.2)	0.0569
I don’t use it	792 (88.4)	(86.3–90.5)	0.8839
**Do you use e-cigarettes containing nicotine?**			
Yes	87 (83.7)	(76.5–90.8)	0.8365
No	17 (16.3)	(9.2–23.5)	0.1635
**How often do you use an e-cigarette during the day?**			
once a day	42 (40.4)	(31.0–49.8)	0.4038
2–5 times a day	16 (15.4)	(8.4–22.3)	0.1538
6–10 times a day	13 (12.5)	(6.1–18.9)	0.125
11–20 times a day	9 (8.6)	(3.3–14.1)	0.0865
more than 20 times a day	24 (23.1)	(15.0–31.2)	0.2308

* *p* < 0.05.

**Table 3 ijerph-19-11584-t003:** Anti-tobacco counseling provided by a family doctor to smoking and non-smoking patients.

Variables	Smoking Status						Total *n* = 896		
	Smoking Patient			Non–Smoking Patient					
	*n* (%) = 190	95% CI	*p* Value	*n* (%) = 706	95% CI	*p* Value	*n* (%) = 896	95% CI	*p* Value
**How often does the family doctor give you advice about tobacco smoking?**									
Never									
Sometimes	60 (31.6)	(25.0–38.2)	0.3158	262 (37.1)	(33.5–40.7)	0.3711	322 (36.0)	(32.8–39.1)	0.3594
Often	97 (51.0)	(43.9–58.2)	0.5105	365 (50.4)	(48.0–55.4)	0.517	453 (50.5)	(47.3–53.8)	0.5056
Always	20 (10.5)	(6.2–14.9)	0.1053	59 (8.4)	(6.3–10.4)	0.0836	79 (8.8)	(7.0–10.7)	0.0882
	13 (6.9)	(3.3–10.4)	0.0684	29 (4.1)	(2.6–5.6)	0.0411	42 (4.7)	(3.3–6.1)	0.0469 *
**Has your family doctor ever advised you to quit smoking?**									
Yes, in the last 12 months									
Yes, in the last 3 months	25 (13.2)	(8.4–18.0)	0.1316	5 (0.7)	(0.09–1.3)	0.0070 **	30 (3.3)	(2.2–4.5)	0.0334 *
Yes, in the last 3 years or more	8 (4.2)	(1.4–7.1)	0.0421 *	2 (0.3)	(–0.1–0.7)	0.0028 **	10 (1.1)	(0.4–1.8)	0.0112 *
No, never	31 (16.3)	(11.1–21.6)	0.1632	16 (2.3)	(1.2–3.4)	0.0227 *	47 (5.2)	(3.8–6.7)	0.0525
I don’t remember	88 (46.3)	(39.2–53.4)	0.4631	627 (88.8)	(86.5–91.1)	0.8881	715 (79.9)	(77.2–82.4)	0.798
	38 (20.0)	(14.3–25.7)	0.2	56 (7.9)	(5.9–9.9)	0.0793	94 (10.5)	(8.5–12.5)	0.1049
**Did your family doctor explain to you the different methods and ways of quitting smoking or provide you with relevant materials?**									
Yes									
No									
	40 (21.1)	(15.3–26.8)	0.2105	9 (1.3)	(0.4–2.1)	0.0127	49 (5.5)	(4.0–7.0)	0.0547
	150 (78.9)	(73.2–84.7)	0.7895	697 (98.7)	(97.9–99.6)	0.9872	847 (94.5)	(93.0–96.0)	0.9453
**Did your family doctor ask you questions about the frequency and quantity of cigarettes smoked?**									
Yes									
No									
	65 (34.2)	(27.5–41.0)	0.3421	37 (5.2)	(3.6–6.9)	0.0524	102 (1.1)	(9.3–13.5)	0.1138
	125 (65.8)	(59.0–72.5)	0.6579	669 (94.8)	(93.1–96.4)	0.9476	794 (88.6)	(86.5–90.7)	0.8862
**Has your family doctor recommended you reduce the number of cigarettes smoked during the day?**									
Yes									
No									
	52 (27.4)	(21.0–33.7)	0.2737	14 (2.0)	(0.9–3.0)	0.0198 *	66 (7.4)	(5.7–9.1)	0.0737
	138 (72.6)	(66.3–79.0)	0.7263	692 (98.0)	(97.0–99.0)	0.9802	830 (92.6)	(90.9–94.3)	0.9263
**Has your family doctor written in your medical records that you smoke?**									
Yes									
No	53 (27.9)	(21.5–34.3)	0.2789	20 (2.8)	(1.6–4.1)	0.0283 *	73 (8.2)	(6.4–9.9)	0.0815
I don’t know	57 (30.0)	(23.5–36.5)	0.3	564 (79.9)	(76.9–82.8)	0.7989	621 (69.3)	(66.3–72.3)	0.693
	80 (42.1)	(35.1–49.1)	0.421	122 (17.3)	(14.5–20.1)	0.1728	202 (22.5)	(19.8–25.3)	0.2254
**Has your family doctor informed you about the negative effects of smoking and the health consequences and the possibility of smoking–related diseases (including cancer)?**									
Yes									
No									
	66 (34.7)	(28.0–41.5)	0.3474	78 (11.0)	(8.7–13.4)	0.1105	144 (16.1)	(13.7–18.5)	0.1607
	124 (65.3)	(58.5–72.0)	0.6526	628 (89.0)	(86.6–91.3)	0.8895	752 (83.9)	(81.5–86.3)	0.8393
**Has your family doctor informed you about the negative impact of smoking on the treatment of your other diseases?**									
Yes									
No									
	56 (29.5)	(23.9–36.0)	0.2947	47 (6.7)	(4.8–8.5)	0.0666	103 (11.5)	(9.4–13.6)	0.1149
	134 (70.5)	(64.0–77.0)	0.7052	659 (93.3)	(91.5–95.2)	0.9334	793 (88.5)	(86.4–90.6)	0.885

* *p* < 0.05; ** *p* < 0.01.

**Table 4 ijerph-19-11584-t004:** Anti-tobacco counseling provided by a family doctor for e-cigarette users and dual users.

Variables	Smoking Status					
	E–Cigarette Users			Dual Users		
	*n* (%) = 104	95% CI	*p* Value	*n* (%) = 65	95% CI	*p* Value
**How often does the family doctor give you advice about smoking/e–cigarette use?**						
Never						
Sometimes	30 (28.9)	(20.1–37.6)	0.2884	18 (27.7)	(16.8–38.6)	0.2769
Often	59 (56.7)	(47.2–66.3)	0.5673	37 (56.9)	(44.9–69.0)	0.5692
Always	8 (7.7)	(2.6–12.8)	0.0769	5 (7.7)	(1.2–14.2)	0.0769
	7 (6.7)	(3.3–10.4)	0.0684	5 (7.7)	(1.2–14.2)	0.0769
**Has your family doctor ever advised you to quit smoking/e–cigarette use?**						
Yes, in the last 12 months						
Yes, in the last 3 months	6 (5.8)	(1.3–10.3)	0.0577	5 (7.7)	(1.2–14.2)	0.0769
Yes, in the last 3 years or more	5 (4.8)	(0.7–8.9)	0.0481 *	3 (4.6)	(–0.5–9.7)	0.0461 *
No, never	7 (6.7)	(1.9–11.5)	0.0673	6 (9.2)	(2.2–16.3)	0.0923
I don’t remember	68 (65.4)	(56.2–74.5)	0.6538	37 (56.9)	(44.9–69.0)	0.5692
	18 (17.3)	(10.0–24.6)	0.1731	14 (21.6)	(11.5–31.5)	0.2154
**Did your family doctor explain to you the different methods and ways of quitting smoking/e–cigarette use or provide you with relevant materials?**						
Yes						
No						
	10 (21.1)	(3.9–15.3)	0.0962	9 (13.8)	(5.4–22.2)	0.1385
	94 (78.9)	(84.7–96.1)	0.9038	56 (86.2)	(77.8–94.6)	0.8615
**Did your family doctor ask you questions about the frequency and amount of e–cigarettes/cigarettes smoked?**						
Yes						
No						
	20 (19.2)	(11.7–26.8)	0.1923	18 (27.7)	(16.8–38.6)	0.2769
	84 (80.8)	(73.2–88.3)	0.8077	47 (72.3)	(61.4–83.2)	0.723
**Has your family doctor recommended you reduce the number of cigarettes or e–cigarettes you smoke during the day?**						
Yes						
No						
	17 (16.3)	(9.2–23.5)	0.1634	13 (20.0)	(10.3–29.7)	0.2
	87 (83.7)	(76.5–90.8)	0.8365	52 (80.0)	(70.3–89.7)	0.8
**Has your family doctor written in your medical records that you use e–cigarettes/ smoke cigarettes?**						
Yes						
No						
I don’t know	18 (17.3)	(10.0–24.6)	0.1731	12 (18.5)	(9.0–27.9)	0.1846
	46 (44.2)	(34.7–53.8)	0.4423	20 (30.7)	(19.5–42.0)	0.3077
	40 (38.5)	(29.1–47.8)	0.3846	33 (50.8)	(38.6–62.9)	0.5077
**Has your family doctor informed you about the negative effects of smoking and the health consequences and the possibility of smoking–related diseases (including cancer)?**						
Yes						
No						
	29 (27.9)	(19.3–36.5)	0.2788	21 (32.3)	(20.9–43.7)	0.3231
	75 (72.1)	(63.5–80.7)	0.7211	44 (67.7)	(56.3–79.1)	0.6769
**Has your family doctor informed you about the negative impact of smoking on the treatment of your other diseases?**						
Yes						
No						
	19 (18.3)	(10.8–25.7)	0.1827	13 (20.0)	(10.3–29.7)	0.2
	85 (81.7)	(74.3–89.2)	0.8173	52 (80.0)	(70.3–89.7)	0.8

* *p* < 0.05.

**Table 5 ijerph-19-11584-t005:** The odds ratio of receiving advice from a family doctor according to the analyzed variables (multivariate logistic regression analysis) in smokers.

Variables	Minimal Anti–Tobacco Intervention					Minimal Anti–Tobacco Intervention for Smokers				
	Total *n* = 574					*n* = 130				
	Unadjusted Model			Adjusted Model		Unadjusted Model			Adjusted Model	
	*n* (%)	OR	95% CI	OR	95% CI	*n* (%)	OR	95% CI	OR	95% CI
**Gender**										
Female	411 (61.8)	1	Ref.	1	Ref.	101 (15.2)	1	Ref.		
Male	163 (70.6)	1.48	(1.07–2.04) **	1.46	(1.03–2.07)	29 (12.6)	0.8	(0.51–1.25)		
**Age (years)**										
<30	155 (60.5)	1	Ref.	1	Ref.	41 (16.0)	1	Ref.		
30–39	84 (68.3)	1.4	(0.89–2.21)	1.5	(0.94–2.40)	21 (17.1)	1.08	(0.61–1.92)		
40–49	144 (67.0)	1.32	(0.90–1.93)	1.23	(0.84–1.81)	28 (13.0)	0.78	(0.47–1.32)		
50–59	59 (56.2)	0.63	(0.39–1.03) *	0.68	(0.42–1.11)	12 (11.4)	0.68	(0.34–1.35)		
60+	132 (67.0)	0.99	(0.66–1.51)	0.85	(0.51–1.39)	28 (14.2)	0.87	(0.51–1.46)		
**Education**										
Primary	15 (57.7)	2.23	(1.00–4.96) *	0.83	(0.35–1.98)	6 (23.1)	2.66	(0.99–7.15) *	0.76	(0.33–1.74)
Medium/Secondary	338 (66.3)	3.21	(2.47–4.12) ***	1.32	0.95–1.82)	84 (16.5)	1.75	(1.12–2.74) **	1.19	(0.88–1.62)
Post–secondary vocational	46 (62.2)	1.02	(0.67–1.54)	1.17	(0.67–2.03)	11 (14.9)	1.54	(0.73–3.27)	0.99	(0.58–1.68)
Higher										
	175 (61.2)	1	Ref.	1	Ref.	29 (10.1)	1	Ref.	1	Ref.
**Marital status**										
Single	252 (64.0)	0.99	(0.74–1.33)			59 (15.0)	1.07	(0.71–1.59)		
Married	240 (64.2)	1	Ref.			53 (14.2)	1	Ref.		
Widowed	47 (68.1)	1.19	(0.69–2.06)			7 (10.1)	0.68	(0.30–1.57)		
Divorced	35 (59.3)	0.81	(0.46–1.43)			11 (18.6)	1.39	(0.68–2.84)		
**Professional situation**										
Unemployed										
Professionally active	31 (68.9)	1.24	(0.64–2.38)			11 (24.4)	1.85	(0.90–3.81)		
Pensioner	354 (64.1)	1	Ref.			82 (14.9)	1	Ref.		
Student/pupil	99 (68.8)	1.23	(0.83–1.82)			16 (11.1)	0.72	(0.41–1.27)		
	90 (58.1)	0.77	(0.54–1.11)			21 (13.5)	0.9	(0.54–1.51)		
**Number of chronic diseases**										
0										
1	262 (62.4)	1	Ref.	1	Ref.	55 (13.1)	1	Ref.	1	Ref.
2	124 (60.5)	0.92	(0.66–1.30)	0.95	(0.67–1.35)	31 (15.1)	1.18	(0.73–1.90)	0.93	(0.66–1.31)
≥ 3	74 (67.9)	1.28	(0.81–1.99)	1.34	(0.83–2.15)	24 (22.0)	1.87	(1.10–3.20) *	1.27	(0.81–1.99)
	114 (70.4)	1.43	(0.97–2.12)*	1.58	(0.97–2.56)	20 (12.3)	0.93	(0.54–1.61)	1.43	(0.96–2.14)
**Alcohol consumption**										
Yes										
No	481 (65.5)	1.41	(0.99–1.99) *	1.45	(1.02–2.07) *	113 (15.4)	1.55	(0.90–2.67)		
	93 (57.4)	1	Ref.	1	Ref.	17 (10.5)	1	Ref.		

* *p* < 0.05, ** *p* < 0.01, *** *p* < 0.001; Fully adjusted model, including all statistically significant characteristics. Ref, reference; CI, confidence interval.

**Table 6 ijerph-19-11584-t006:** The odds ratio of receiving advice from a family doctor according to the analyzed variables (multivariate logistic regression analysis) in e-cigarette users and dual users.

Variables	Minimal Anti–Tobacco Intervention for E–Cigarette Users *n* = 74					Minimal Anti–Tobacco Intervention For Dual Users *n* = 47				
	Unadjusted Model			Adjusted Model		Unadjusted Model			Adjusted Model	
	*n* (%)	OR	95% Cl	OR	95% Cl	*n* (%)	OR	95% Cl	OR	95% Cl
**Gender**										
Female	58 (8.7)	1	Ref.			36 (5.4)	1	Ref.		
Male	16 (6.9)	0.78	(0.44–1.38)			11 (4.8)	0.87	(0.44–1.75)		
**Age (years)**										
<30	33 (12.9)	1	Ref.	1	Ref.	19 (7.4)	1	Ref.	1	Ref.
30–39	7 (5.7)	0.41	(0.17–0.95) *	0.48	(0.19–1.23)	7 (5.7)	0.75	(0.29–1.81)	1	(0.36–2.83)
40–49	17 (6.8)	0.58	(0.31–1.07) *	0.57	(0.29–1.13)	10 (4.7)	0.61	(0.28–1.34)	0.6	(0.25–1.44)
50–59	9 (8.6)	0.63	(0.29–1.38)	0.61	(0.23–1.63)	4 (3.8)	0.49	(0.16–1.49)	0.52	(0.14–1.99)
60+	8 (4.1)	0.29	(0.13–0.63) ***	0.37	(0.11–1.28)	7 (3.6)	0.46	(0.19–1.12) *	0.75	(0.19–2.97)
**Education**										
Primary	1 (3.8)	0.63	(0.08–4.96)	1.07	(0.13–9.04)	1 (3.8)	1.23	(0.15–10.11)	1.85	(0.20–17.09)
Medium/Secondary	51 (10.0)	1.76	(0.99–3.11) *	2.06	(1.12–3.75) *	33 (6.5)	2.13	(1.00–4.52) *	2.51	(1.12–5.64) *
Post–secondary vocational										
Higher	5 (6.8)	1.15	(0.41–3.22)	1.46	(0.51–4.22)	4 (5.4)	1.76	(0.53–5.89)	2.05	(0.59–7.10)
	17 (5.9)	1	Ref.	1	Ref.	9 (3.1)	1	Ref.	1	Ref.
**Marital status**										
Single	43 (10.9)	2.17	(1.25–3.76) **	1.32	(0.63–2.78)	26 (6.6)	1.96	(0.99–3.88) *	1.19	(0.47–2.99)
Married	20 (5.3)	1	Ref.	1	Ref.	13 (3.5)	1	Ref.	1	Ref.
Widowed	5 (7.2)	1.38	(0.50–3.82)	2.79	(0.83–9.38)	3 (4.3)	1.26	(0.35–4.55)	1.93	(0.44–8.43)
Divorced	6 (10.2)	2	(0.77–5.22)	1.78	(0.67–4.75)	5 (8.5)	2.57	(0.88–7.50)	2.38	(0.79–7.17)
**Professional situation**										
Unemployed										
Professionally active	5 (11.1)	1.28	(0.43–3.23)	1.15	(0.42–3.14)	4 (8.9)	4.59	(0.99–21.32) *	1.69	(0.55–5.22)
Pensioner	49 (8.9)	1	Ref.	1	Ref.	28 (5.1)	1	Ref.	1	Ref.
Student/ pupil	4 (2.8)	0.29	(0.10–0.83) ***	0.31	(0.08–1.22)	3 (2.1)	2.51	(0.75–8.38)	0.28	(0.06–1.33)
	16 (10.3)	1.66	(0.91–3.04)	0.7	(0.35–1.38)	12 (7.7)	3.94	(1.09–14.27) *	1.15	(0.50–5.22)
**Number of chronic diseases**										
0										
1	38 (9.0)	1	Ref.			21 (5.0)	1	Ref.		
2	17 (8.3)	0.91	(0.50–1.65)			12 (5.9)	1.18	(0.57–2.45)		
≥ 3	9 (8.3)	0.9	(0.42–1.93)			6 (5.5)	1.11	(0.44–2.81)		
	10 (6.2)	0.66	(0.32–1.36)			8 (4.9)	0.99	(0.43–2.28)		
**Alcohol consumption**										
Yes										
No	62 (8.4)	1.15	(0.61–2.19)			39 (5.3)	1.08	(0.50–2.36)		
	12 (7.4)	1	Ref.			8 (4.9)	1	Ref.		

* *p* < 0.05, ** *p* < 0.01, *** *p* < 0.001; Fully adjusted model, including all statistically significant characteristics. Ref, reference; CI, confidence interval.

## Data Availability

The datasets used and/or analyzed during the current study are available from the corresponding author upon reasonable request.

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
