# Peer review of "The Frequency of Tobacco Smoking and E-Cigarettes Use among Primary Health Care Patients—The Association between Anti-Tobacco Interventions and Smoking in Poland"

_ijerph, 2022, doi:10.3390/ijerph191811584_

Round 1
Reviewer 1 Report
The study assess the prevalence of smoking and e-cigarettes use among primary care patients during the COVID-19 pandemic and to assess the frequency of minimal anti-tobacco interventions by family doctors in patients. It's well organized study and only minor comments from me.
Comment 1
The study investigated the association between frequency of smoking and intervention. The "and" in the title could be changed into "The association of anti-tobacco interventions and smoking", or different word with similar meanings.
Comment 2
The introduction has to many paragraphs. Some of them can be integrated to strengthen the point.
Comment 3
Is there any results that can be shown in a figure? Figures are much easier to understand.
Author Response
Response to Reviewer 1 Comments
We have carefully considered all reviewer's considerations of the paper. Please find enclosed our detailed answers to reviewer's queries.
Point 1:
The study investigated the association between frequency of smoking and intervention. The "and" in the title could be changed into "The association of anti-tobacco interventions and smoking", or different word with similar meanings.
Response 1: The authors really appreciate all your kindly comments. We changed "and" in the title into "The association of anti-tobacco interventions and smoking"
Point 2:
The introduction has to many paragraphs. Some of them can be integrated to strengthen the point.
Response 2: Thank you for your comment. The text of the introduction was modified according to your proposal.
Point 3:
Is there any results that can be shown in a figure? Figures are much easier to understand.
Response 3:
Thank you for your valuable comments. We added a figure 1: ,,The prevalence of daily tobacco smoking” and a figure 2 ,, The prevalence of daily use of e-cigarettes”.
Table S1 and Table S2 in Supplementary materials
The text was checked by a native speaker of English.
Reviewer 2 Report
Dear Authors,
Good study. Please see the comments below:
1) Minor spelling or grammatical check required.
2) Overall good methodology followed and developed to address an problem which will have significant impact in lives of people.
3) Study demonstrates the lack of education in primary care facilities which is the first contact of care for patients. Proper training of doctors in primary care facilites can have bigger impact overall and this study explains that well.
Author Response
Response to Reviewer 2 Comments
We have carefully considered all reviewer's considerations of the paper. Please find enclosed our detailed answers to reviewer's queries.
Point 1:
Minor spelling or grammatical check required.
Response 1:
The text was checked by a native speaker of English.
Point 2:
Overall good methodology followed and developed to address an problem which will have significant impact in lives of people.
Response 2: Thank you for your comment and good feedback on our manuscript.
Point 3:
Study demonstrates the lack of education in primary care facilities which is the first contact of care for patients. Proper training of doctors in primary care facilites can have bigger impact overall and this study explains that well.
Response 3:
Thank you for your comment. The text was modified according to your proposal. We added ,,Our findings indicate a lack of education in primary care facilities, which represent the first point of contact for patients. They also demonstrate that proper training of staff in this area can have a considerable overall impact”.
Reviewer 3 Report
This is a very worthwhile report on both tobacco and e-cigarette use in Poland, and the levels of advice and support provided by family doctors.
The paper appropriately recognises that the study was carried out in the context of broader developments, ranging from the expansion and promotion of e-cigarette use to the COVID pandemic. Nonetheless, the findings are of interest both for Poland and more broadly.
For readers unfamiliar with the Polish family practice system (access, use, costs, etc.), it might be useful to provide some information on this either in the text or as a footnote.
The study appears to have been well conducted, and is clearly reported. There may be a need for some minor editorial amendments (e.g. p5 “24.7 of menthol cigarettes”). This reviewer was also unfamiliar with the term “hand twisted” – does this refer to hand-rolled cigarettes?
The paper reports continuing fairly high smoking prevalence, alongside the growing phenomenon of e-cigarette use that has developed in recent years. It rightly notes that despite decades of evidence on the benefits of even brief advice by general (/family) practitioners, such advice is clearly not occurring to anything even approaching the extent that it should. This is sadly not surprising, but is disturbing, and should be a matter of concern to both the government and medical organisations.
The sections on actions required from government could be strengthened. The paper notes that “In Poland, preventive actions are scattered and uncoordinated, not only in theCOVID-19 pandemic. There is no strategy about what such an intervention should look like, and to who it should be addressed. Preventive measures are underfunded, the organization is bad, the staff is misused, and there is no evaluation of the activities carried out.” This is well put, but the paper then goes on to focus on lung cancer screening. It might be better and more appropriate here to include a call for strong and comprehensive action by government and health authorities to implement the comprehensive approach recommended by WHO. This would include the measures set out in the WHO Framework Convention on Tobacco Control (FCTC) – including Article 14, which outlines best practice approaches to smoking cessation, as well as the other measures (strong curbs on tobacco industry marketing and other activity, strong health warnings, price/tax policy, public education, etc.) that are required to reduce smoking in the community.
It might also be worth noting that effective action on tobacco by governments around the world has generally followed calls for action by medical and health organisations; so there may well be a role for such approaches in Poland.
The authors note that “Primary care physicians should try to improve their qualifications in this area, which will increase the effectiveness of MAI”. It might also be useful to add a comment about the onus on those who train primary care physicians to ensure that smoking cessation is given appropriate emphasis in the curriculum.
The authors have done well to focus attention on an important area, and it is to be hoped that their excellent work will result in a greater focus on action to reduce smoking in the community at large, and by family doctors in particular.
Author Response
Response to Reviewer 3 Comments
We have carefully considered all reviewer's considerations of the paper. Please find enclosed our detailed answers to reviewer's queries.
Point 1:
For readers unfamiliar with the Polish family practice system (access, use, costs, etc.), it might be useful to provide some information on this either in the text or as a footnote.
Response 1: The authors really appreciate all your kindly comments. We added information about the Polish family practice system (access, use, costs) in the discussion and as a footnote.
Point 2:
The study appears to have been well conducted, and is clearly reported. There may be a need for some minor editorial amendments (e.g. p5 “24.7 of menthol cigarettes”). This reviewer was also unfamiliar with the term “hand twisted” – does this refer to hand-rolled cigarettes?
Response 2: Thank you for your comment. We corrected it.
p5 “24.7 of menthol cigarettes”, we changed to 24.7% menthol cigarettes. The term ‘’hand twisted’’ refer to ,,hand rolled cigarettes”, we changed this term.
Point 3:
The sections on actions required from government could be strengthened. The paper notes that “In Poland, preventive actions are scattered and uncoordinated, not only in theCOVID-19 pandemic. There is no strategy about what such an intervention should look like, and to who it should be addressed. Preventive measures are underfunded, the organization is bad, the staff is misused, and there is no evaluation of the activities carried out.” This is well put, but the paper then goes on to focus on lung cancer screening. It might be better and more appropriate here to include a call for strong and comprehensive action by government and health authorities to implement the comprehensive approach recommended by WHO. This would include the measures set out in the WHO Framework Convention on Tobacco Control (FCTC) – including Article 14, which outlines best practice approaches to smoking cessation, as well as the other measures (strong curbs on tobacco industry marketing and other activity, strong health warnings, price/tax policy, public education, etc.) that are required to reduce smoking in the community. It might also be worth noting that effective action on tobacco by governments around the world has generally followed calls for action by medical and health organisations; so there may well be a role for such approaches in Poland.
Response 3:
We corrected it.
Point 4:
The authors note that “Primary care physicians should try to improve their qualifications in this area, which will increase the effectiveness of MAI”. It might also be useful to add a comment about the onus on those who train primary care physicians to ensure that smoking cessation is given appropriate emphasis in the curriculum.
The authors have done well to focus attention on an important area, and it is to be hoped that their excellent work will result in a greater focus on action to reduce smoking in the community at large, and by family doctors in particular.
Response 4:
We corrected it.
The text was checked by a native speaker of English